# Seroepidemiologic survey of emerging vector-borne infections in South Korean forest/field workers

Ji Yun Noh[1], Joon Young Song[1]*, Joon Yong Bae[2], Man-Seong Park[2], Jin Gu Yoon[1], Hee Jin Cheong[1], Woo Joo Kim[1]

1 Division of Infectious Diseases, Department of Internal Medicine, Korea University Guro Hospital, Korea University College of Medicine, Seoul, Republic of Korea, 2 Department of Microbiology, Institute for Viral Diseases, Korea University College of Medicine, Seoul, Republic of Korea

* infection@korea.ac.kr

**Data Availability Statement:** All relevant data are within the manuscript and its Supporting Information files.

## Abstract

With global warming and lush forest change, vector-borne infections are expected to increase in the number and diversity of agents. Since the first report of severe fever with thrombocytopenia syndrome (SFTS) in 2013, the number of reported cases has increased annually in South Korea. However, although tick-borne encephalitis virus (TBEV) was detected from ticks and wild rodents, there is no human TBE case report in South Korea. This study aimed to determine the seroprevalence of TBEV and SFTS virus (SFTSV) among forest and field workers in South Korea. From January 2017 to August 2018, a total 583 sera were obtained from the forest and field workers in South Korea. IgG enzyme-linked immunosorbent assay (ELISA) and neutralization assay were conducted for TBEV, and indirect immunofluorescence assay (IFA) and neutralization assay were performed for SFTSV. Seroprevalence of TBEV was 0.9% (5/583) by IgG ELISA, and 0.3% (2/583) by neutralization assay. Neutralizing antibody against TBEV was detected in a forest worker in Jeju (1:113) and Hongcheon (1:10). Only 1 (0.2%) forest worker in Yeongju was seropositive for SFTSV by IFA (1:2,048) and neutralizing antibody was detected also. In conclusion, this study shows that it is necessary to raise the awareness of physicians about TBEV infection and to make efforts to survey and diagnose vector-borne diseases in South Korea.

## Author summary

With global warming and lush forest change, vector-borne infections are expected to increase in the number and diversity of agents. Since the first report of severe fever with thrombocytopenia syndrome (SFTS) in 2013, the number of reported cases has increased annually in South Korea. However, although tick-borne encephalitis virus (TBEV) was detected from ticks and wild rodents, there is no human TBE case report in South Korea. This study aimed to determine the seroprevalence of TBEV and SFTS virus (SFTSV) among forest and field workers in South Korea. Among 583 forest/field workers, the seroprevalence of neutralizing antibodies against TBEV and SFTSV were 0.3% (2/583) and

**Funding:** JYS was supported by a Korea University Guro Hospital grant (No. I1602131) that was underwritten by Pfizer. The funders had no role in study design, data collection and analysis, decision to publish, or preparation of the manuscript.

**Competing interests:** I have read the journal's policy and the authors of this manuscript have the following competing interests: JYS received a grant that was underwritten by Pfizer. The other authors declare that no competing interests exist.

0.2% (1/583), respectively. This study shows that it is necessary to raise the awareness of physicians about TBEV infection and to make efforts to survey and diagnose vector-borne diseases in South Korea.

## Introduction

In South Korea, several new tick-borne infectious diseases have been identified since the 2000s, including severe fever with thrombocytopenia syndrome (SFTS), anaplasmosis, ehrlichiosis, and bartonellosis [1–4]. Although existed long time, SFTS was first notified in 2013, and showed the alarming increase in the number of annually reported cases in South Korea. Furthermore, in neighboring mainland China, more than 30 emerging tick-associated infectious agents have been reported to cause human infection [5]. Similar to SFTS, some vector-borne diseases might be missed without clinical suspicion. Tick-borne encephalitis virus (TBEV) has been an important causative agent of viral infection of the central nervous system in Europe, Russia, northern China and Japan [6]. In South Korea, TBEV was detected from ticks and wild rodents, however, no human cases of TBE have been reported so far [7].

With global warming and lush forest change, vector-borne infections are expected to steadily increase with respect to the number and diversity of agents. Thus, we conducted this study to determine the seroprevalence of TBEV and SFTS virus (SFTSV) among forest and field workers in South Korea. Seroepidemiology for West Nile virus (WNV), dengue virus (DENV) and Japanese encephalitis virus (JEV) was also tested, because they belong to genus *Flavivirus* along with TBEV. We conducted this serosurveillance to find some evidence of emerging vector-borne infectious diseases in South Korea.

## Methods

From January 2017 to August 2018, a total 583 sera were obtained from the forest and field workers in South Korea (Fig 1). Sites for sample collection were selected based on previous reports on the detection of TBEV in tick in South Korea and feasibility to access forest and field workers for the study. Data on vaccination history, past history of vector-borne diseases, and tick-bite experience were investigated based on the self-report by study subjects. The study protocol was approved by the Institutional Review Board of Korea University Guro Hospital (approval number: 2016GR0173). All participants gave written informed consent.

Anti-TBEV IgG was evaluated using the Anti-TBE Virus ELISA (IgG) (EI 2661-9601G, Euroimmune, Germany) according to the manufacturer's protocol. Human IgG against WNV, DENV, and JEV which could be cross-reactive to TBEV were measured using the Anti-West Nile Virus ELISA (EI 2662-9601G, Euroimmune), Anti-Dengue Virus ELISA (EI 266b-9601G, Euroimmune), and Anti-JEV ELISA (EI 2663-9601G, Euroimmune), respectively. According to the manufacturer's guide, ELISA results were interpreted as follows: < 16 relative units (RU)/mL, negative; $\geq$ 16 to < 22 RU/mL, borderline; $\geq$ 22 RU/mL, positive. Samples which revealed as borderline or positive were examined repeatedly, and the result was designated according to the average of two values by ELISA. Finally, positive results were considered as seropositivity, and borderline results were regarded as negative.

Neutralization assay for TBEV was performed as described previously [8]. Briefly, serial dilutions of sera were incubated with approximately 100 tissue culture infective doses (TCID) of TBEV. A Western (European) subtype of TBEV, Neudörfl strain was used in this assay. Replicates of the mixtures were incubated for 7 days on TBEV-susceptible Vero cells seeded in 96-well microtiter plates. The resulting supernatants were tested for presence of TBEV by a

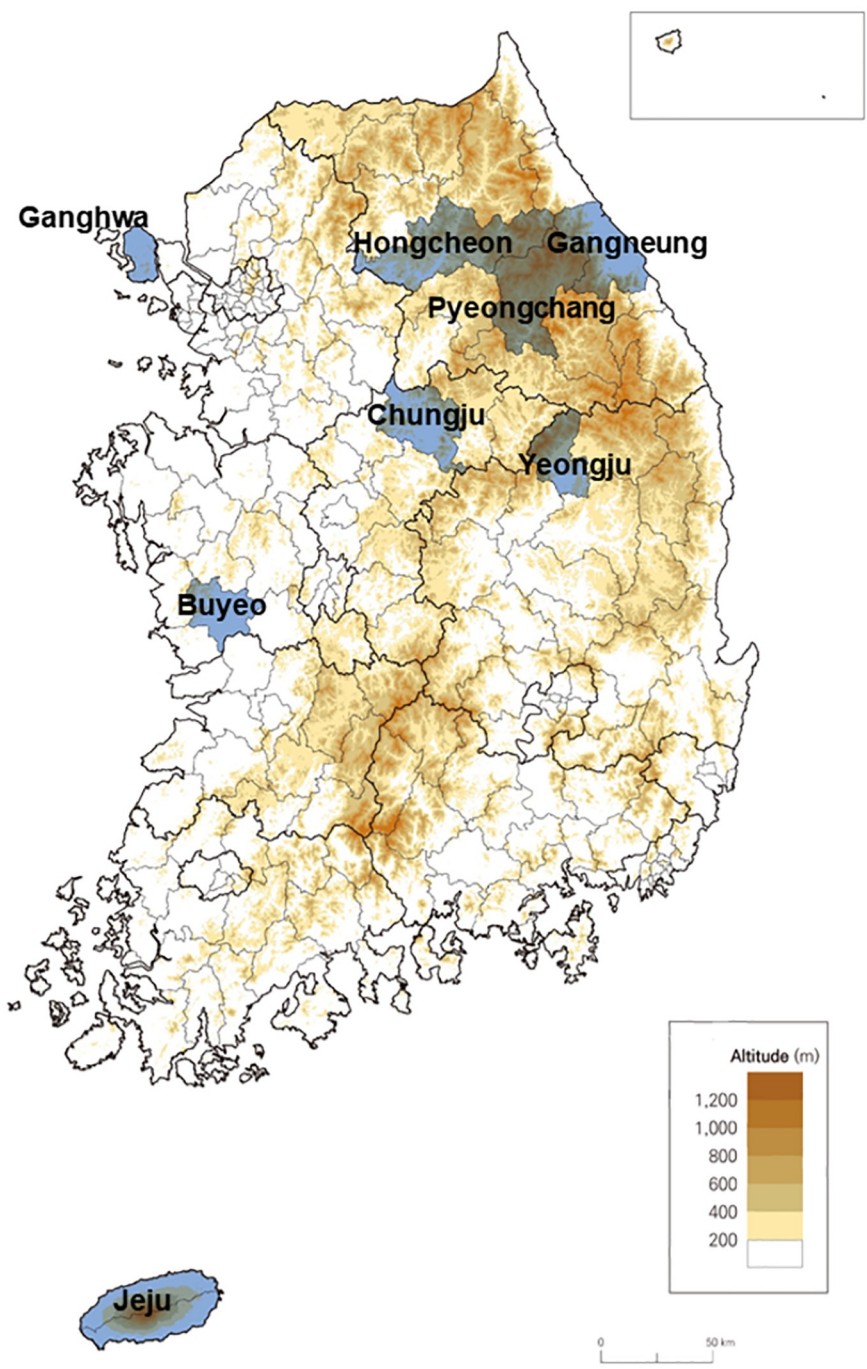

**Fig 1. Map of South Korea.** Sites where blood sampling was conducted from the forest and field workers were presented with blue color. The map in Fig 1 was drawn by the researchers and did not use a base layer.

four-layer ELISA as described previously [9]. The sample dilution resulting in virus neutralization in 50% of the replicates (NT50) was calculated using the method of Spearman and Karber [10,11]. A cut-off value was set to 0.05 based on the titration of a known concentration of TBE viral antigen. Sample with ≥ 1:10 titer for neutralization assay was interpreted as a positive result.

Anti-SFTSV IgG was examined using indirect immunofluorescence assay (IFA). Antigen slides of Vero E6 cells infected with SFTSV were prepared. SFTSV isolated from a PCR-confirmed SFTS patient in South Korea was used. Sera were diluted in 2-fold serial dilutions starting 1:32 to 1:4,096. Sera were placed onto aceton-fixed well slides and reacted at 37˚C for 30 min. After incubation, the slides were washed with phosphate-buffered saline and dextrose water. Fluorescein isothiocyanate (FITC)-conjugated anti-human IgG (109-095-003, Jackson immune research, USA) was added to each well and the slides were incubated at 37˚C for 30 min. After washing, SFTSV-specific fluorescence was examined using fluorescent microscope (EVOS FL, Thermo fisher scientific, USA).

Plaque reduction neutralization test (PRNT) for SFTSV was performed using clinical isolate of SFTSV. Positive sample for anti-SFTSV IgG by IFA was tested. Briefly, sera were diluted in 2-fold serial dilutions from 1:10 to 1:2,560 using 2x EMEM (Lonza, Switzerland) with 1% gentamicin sulfate. Approximately 100 plaque-forming units of SFTSV were mixed with serially diluted sera samples. After 1 hr incubation at 37˚C, Vero E6 cells were inoculated with each virus-serum mixture for 2 hr at 37˚C. The inoculum was discarded and cells were overlaid with 3 mL of first 2x EMEM overlay media containing 0.8% agarose. After 5 days of incubation at 37˚C, second 2 mL of 2x EMEM overlay media containing 0.8% agarose with 5% neutral red solution were supplemented. After 2 days, PRNT titer of each serum was determined with the highest serum dilution inhibiting more than 50% viral plaques compared with the mean plaque count of virus control.

## Results

Among 583 study subjects, male was 503 (86.3%) and median age was 56 years. Forest workers was 552 (94.7%), and 57 (9.8%) worked at field. Among study population, 26 worked at both forest and field. Duration of career was available in 99.3% (548/552) of forest workers and 100% of field workers. The median duration of career was 10 years in forest workers and 24 years in field workers. Only 19 (3.3%) reported experience of tick bite previously (Table 1). No subjects were immunized with yellow fever vaccine. All subjects denied any history of illness due to dengue fever, yellow fever, and Japanese encephalitis.

Seroprevalence of anti-TBEV IgG was 0.9% (5/583, 95% confidence interval [CI], 0.3–2.1) by ELISA (Table 2). Neutralization assay for TBEV showed positive rate of 0.3% (2/583, 95% CI, 0.1–1.4). One forest worker in Jeju had neutralizing antibody against TBEV; anti-TBEV IgG titer measured using ELISA was 56.1 RU/mL, and neutralization titer was 1:113. This man has been working as a forest worker for 6 years. Anti-DENV IgG, anti-WNV IgG, and anti-JEV IgG were negative in this man. The other forest worker in Hongcheon showed 1:10 of neutralization titer against TBEV and his duration of career was 25 years. However, anti-TBEV IgG was not detected using ELISA in this person.

Among 583 subjects, positive rates for anti-DENV IgG and anti-WNV IgG were 0.5% (95% CI, 0.1–1.6)(Table 2). Peoples who have anti-DENV IgG or anti-WNV IgG were also positive for anti-JEV IgG. Overall positive rate of anti-JEV IgG tested using ELISA was 15.8% (95% CI, 13.0–19.1). Only 1 forest worker in Yeongju was seropositive for SFTSV by IFA (Fig 2); anti-SFTSV antibody titer was measured as 1:2,048. Neutralizing antibody titer against SFTSV was 1:80. Anti-TBEV IgG, anti-DENV IgG, anti-WNV IgG, and anti-JEV IgG were negative in this forest worker. Among 19 persons with tick-bite experience, three (15.8%) were positive for anti-JEV IgG and all were negative for anti-TBEV IgG, anti-DENV IgG, anti-WNV IgG, and anti-SFTSV IgG.

## Discussion

In this study, seroprevalence of anti-TBEV IgG was 0.9% (95% CI, 0.3–2.1) by ELISA and 0.3% (95% CI, 0.1–1.4) by neutralization assay among South Korean forest and field workers.

**Table 1. Demographic characteristics of the study population.**

| | Value (n = 583) |
|---|---|
| Sex (male), n (%) | 503 (86.3) |
| Age* (year) | 56 (19–88) |
| 19 | 3 (0.5) |
| 20–29 | 37 (6.3) |
| 30–39 | 54 (9.3) |
| 40–49 | 80 (13.7) |
| 50–59 | 192 (32.9) |
| 60–69 | 168 (28.8) |
| 70–79 | 40 (6.9) |
| 80–89 | 9 (1.5) |
| Site | |
| Ganghwa | 18 (3.1) |
| Hongcheon | 113 (19.4) |
| Pyeongchang | 43 (7.4) |
| Gangneung | 118 (20.2) |
| Chungju | 25 (4.3) |
| Buyeo | 78 (13.4) |
| Yeongju | 71 (12.2) |
| Jeju | 117 (20.1) |
| Sampling year | |
| 2017 | 240 (41.2) |
| 2018 | 343 (58.8) |
| Occupation | |
| Forest worker | 552 (94.7) |
| Duration* (year) | 10 (0.08–50) |
| Field worker | 57 (9.8) |
| Duration* (year) | 24 (1–60) |
| Vaccination history | |
| Japanese encephalitis virus | 7 (1.2) |
| Yellow fever virus | 0 (0) |
| Past history of vector-borne disease | |
| Japanese encephalitis | 0 (0) |
| Yellow fever | 0 (0) |
| Dengue | 0 (0) |
| Self-report tick-bite experience | 19 (3.3) |

*median (range)

Although no TBE case has been reported in South Korea yet, there is a chance that TBE cases might be overlooked in the clinical setting due to the low awareness of the disease and limited diagnostic methods. In South Korea, several field surveillances revealed wide distribution of TBEV in ticks and TBEV strains belonged to the Western subtype [12–15]. The minimal infection rates with TBEV for *Haemaphysalis longicornis*, *Haemaphysalis flava*, and *Ixodes nipponensis* were 0.06%, 0.17%, and 2.38% [7]. TBEV is an important viral agent of the central nervous system infections in European countries, northern China, Mongolia, and Russia [16]. Approximately 10,000–12,000 cases of TBE are reported annually, but the actual incidence of TBE may be underestimated [16]. Recently, tick-borne infections are rising and various factors

**Table 2. Seroprevalence of TBEV, WNV, DENV, JEV, and SFTSV in South Korean forest/field workers.**

| | Number of seropositive subjects (%) | | | | | | | | |
|---|---|---|---|---|---|---|---|---|---|
| | Ganghwa (n = 18) | Hongcheon (n = 113) | Pyeongchang (n = 43) | Gangneung (n = 118) | Chungju (n = 25) | Buyeo (n = 78) | Yeongju (n = 71) | Jeju (n = 117) | Total (n = 583) |
| TBEV[a] | - | - | - | 3 (2.5) | - | - | - | 2 (1.7) | 5 (0.9)[c] |
| WNV[a] | - | - | - | 1 (0.8) | - | - | 1 (1.4) | 1 (0.9) | 3 (0.5) |
| DENV[a] | - | - | - | 2 (1.7) | - | - | - | 1 (0.9) | 3 (0.5) |
| JEV[a] | 6 (33.3) | 10 (8.8) | 3 (7.0) | 29 (24.6) | 4 (16.0) | 15 (19.2) | 7 (9.9) | 18 (15.4) | 92 (15.8) |
| SFTSV[b] | - | - | - | - | - | - | 1 (1.4) | - | 1 (0.2) |

TBEV, tick-borne encephalitis virus; WNV, West Nile virus; DENV, Dengue virus; JEV, Japanese encephalitis virus; SFTSV, severe fever with thrombocytopenia syndrome virus

a: examined by ELISA

b. examined by immunofluorescence assay

c: Among 5 samples, only 1 sample from Jeju was positive for antibody against TBEV in neutralization test.

such as global warming, increased outdoor activities and increased overseas travel and trade may contribute to the expansion of tick-borne infections in humans.

Compared to the results in this study, previous serosurvey among farmers in Jeju, South Korea during 2015–2018 reported rather higher seroprevalence (1.3%) of anti-TBE-IgG by ELISA [17]. Most serological assays for TBEV can detect cross-reactive antibodies against other viruses of genus *Flavivirus* [18]. Thus, neutralization test is considered as a standard assay detecting the specific antibodies against TBEV. In this study, neutralizing antibody against TBEV was detected in a forest worker in Jeju and Hongcheon. Previous studies showed detection of TBEV from ticks in Jeju [14,15]. Thus, there is a possibility that a seropositive person in Jeju was infected with TBEV in South Korea. However, he immigrated to Jeju from Jilin in northern China 8 years ago, therefore, TBEV infection could have occurred when he lived in China. Seropositive rate of TBEV among healthy people was 0%-10.9% in previous studies which were conducted at 3 areas including Jilin in China [19–21]. Nevertheless, the seropositive person in the present study might have a chance to be exposed to TBEV at Jeju island considering the remarkably high antibody titer (anti-TBEV IgG 56.1 RU/mL; neutralization titer 113). Although the pathogenesis and immune responses of TBEV infection are not fully

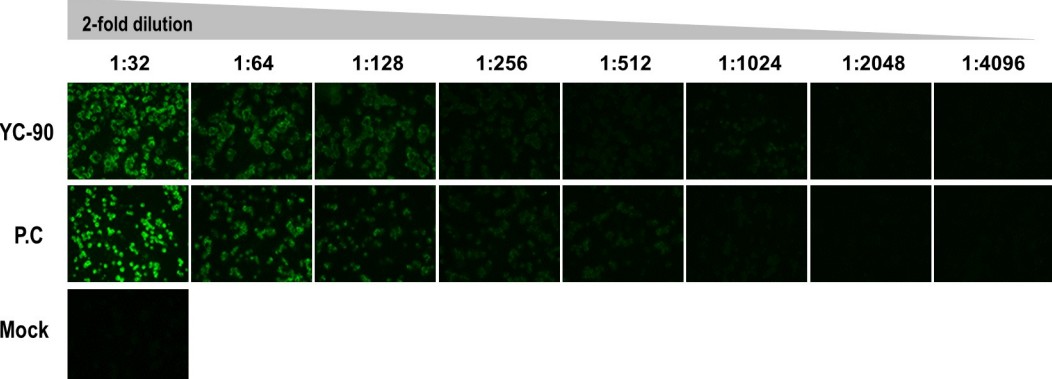

**Fig 2. Detection of anti-SFTSV IgG using indirect immunofluorescence assay in serum of a forest worker; YC-90 (forest worker serum), P.C (serum from a PCR-confirmed SFTS patient), and mock (serum from a healthy donor without any history of SFTS).**

elucidated, TBEV-specific antibody response is known to reach peak in the 6-week after infection and decrease to a lower level following a period of up to 1 year [22,23].

Since the first report of SFTS in South Korea, the number of reported cases has increased annually; more than 200 cases per year were reported in 2017 and 2018 [1,24]. Persons who work in forest or field and enjoy outdoor activities are considered to have more frequent opportunity to be exposed to ticks. The infection rate of SFTSV in adult and nymph ticks was 3.67% (54/1,470) in five national parks in South Korea [25]. In this study, seroprevalence of SFTSV was 0.2% (1/583) by IFA among forest and field workers. Previous study showed 4.1% of seropositivity of SFTSV in residents of 3 rural areas in 2014 in South Korea [26]. In the patients who visit a tertiary hospital for various reasons, SFTSV seroprevalence was revealed as 2.1% by ELISA and seropositive rate was higher in patients from rural area than those from urban area [27]. Seroprevalence could be varied depending on study population, study site and time, and laboratory methods. In a study of serosurveillance for SFTSV antibodies which was conducted in Japan, among 694 samples, 8 (1.2%) showed positive reactions in ELISA. However, when these positive samples were further assessed with IFA and neutralization assay, seropositive rate decreased; 2 (0.3%) samples were positive by IFA and 1 (0.1%) sample by neutralization assay [28]. This suggests that seroprevalence of SFTSV measured with ELISA can be overestimated owing to possible nonspecific reactions.

In this study, we analyzed anti-JEV IgG because of endemicity of JEV in South Korea and its possible cross-reactivities to TBEV and other flaviviruses. Japanese encephalitis virus, a mosquito-borne flavivirus, is endemic in South Korea. In a study on seroprevalence of neutralizing antibodies against JEV among the general population in South Korea, the seropositive rate was 97%-99% for the age of 20–29 years, 80%-89% in 30–49 years, and 75%-80% in 50–69 years [29]. Peoples who have anti-DENV IgG or anti-WNV IgG were also positive for anti-JEV IgG in our study. Thus, it is possible that cross-reactivities of anti-JEV antibodies to DENV or WNV contribute to the seropositivity of anti-DENV IgG and anti-WNV IgG. However, it is not definite because we did not perform neutralization assays and PCR for DENV, WNV and JEV.

This study has some limitations. First, one of two positive sera by neutralization assay against TBEV was negative when tested for anti-TBEV IgG using ELISA. The neutralizing antibody titer was equal to the cut-off value (1:10). Assay for anti-TBEV IgM and IFA were not conducted in this study. Second, PCR was not conducted in this study because the study population was not in acute phase of suspected infection and this study aimed to investigate seroprevalence of forest/field workers. Third, only 19 subjects reported any experience of tick bite, however, there is a possibility that tick bite was unrecognized or neglected.

However, this study is valuable because neutralization assay of TBEV was performed to investigate seroprevalence and a serosurvey was performed at forest and field workers in several area of South Korea. In this study, antibodies against WNV, DENV, and JEV were evaluated simultaneously which belong to genus *Flavivirus* same as TBEV [30]. Among 4 samples which is positive for anti-TBEV IgG in ELISA and negative for neutralizing antibody against TBEV, all samples were negative for antibodies for both DENV and WNV, and anti-JEV IgG with a range of borderline was detected in 1 sample. Due to the interference by cross-reactive antibodies of other flaviviruses, neutralization test is necessary to confirm positive ELISA results to assess immunity, especially in areas where TBEV infection is not endemic [31].

In conclusion, seroprevalence of neutralizing antibody for TBEV was 0.3% and seropositive rate of antibody for SFTSV was 0.2% among South Korean forest and field workers. This study shows that it is necessary to raise the awareness of physicians about TBEV infection and to make efforts to survey and diagnose vector-borne diseases in South Korea. Threats of vector-

borne infections to public health may continue. Serological tests for neglected vector-borne diseases need to be standardized for the accurate diagnosis.

## Acknowledgments

Authors thank Bogdan Jug in QC/Biolab, Pfizer Manufacturing Austria for the neutralization testing against TBEV.

## Author Contributions

**Conceptualization:** Joon Young Song.

**Data curation:** Joon Young Song, Jin Gu Yoon, Hee Jin Cheong, Woo Joo Kim.

**Formal analysis:** Ji Yun Noh.

**Funding acquisition:** Joon Young Song.

**Investigation:** Ji Yun Noh, Joon Young Song, Jin Gu Yoon, Hee Jin Cheong, Woo Joo Kim.

**Methodology:** Joon Young Song, Joon Yong Bae, Man-Seong Park.

**Project administration:** Joon Young Song.

**Resources:** Ji Yun Noh, Joon Young Song, Joon Yong Bae, Man-Seong Park.

**Software:** Joon Young Song.

**Supervision:** Joon Young Song.

**Validation:** Ji Yun Noh, Joon Young Song, Joon Yong Bae, Man-Seong Park, Jin Gu Yoon, Woo Joo Kim.

**Writing – original draft:** Ji Yun Noh, Joon Young Song.

**Writing – review & editing:** Ji Yun Noh, Joon Young Song, Joon Yong Bae, Man-Seong Park, Jin Gu Yoon, Hee Jin Cheong, Woo Joo Kim.

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
