## [Decision Letter · Decision Letter 0]

6 Mar 2021

Dear Dr. Song,

Thank you very much for submitting your manuscript "Seroepidemiologic survey of emerging vector-borne infections in South Korean forest/field workers" for consideration at PLOS Neglected Tropical Diseases. As with all papers reviewed by the journal, your manuscript was reviewed by members of the editorial board and by several independent reviewers. In light of the reviews (below this email), we would like to invite the resubmission of a significantly-revised version that takes into account the reviewers' comments. 

We cannot make any decision about publication until we have seen the revised manuscript and your response to the reviewers' comments. Your revised manuscript is also likely to be sent to reviewers for further evaluation.

Sincerely,

Nam-Hyuk Cho

Deputy Editor

Nam-Hyuk Cho

Deputy Editor

Reviewer's Responses to Questions

**Key Review Criteria Required for Acceptance?**

**Methods**

-Are the objectives of the study clearly articulated with a clear testable hypothesis stated?

-Is the study design appropriate to address the stated objectives?

-Is the population clearly described and appropriate for the hypothesis being tested?

-Is the sample size sufficient to ensure adequate power to address the hypothesis being tested?

-Were correct statistical analysis used to support conclusions?

-Are there concerns about ethical or regulatory requirements being met?

Reviewer #1: (No Response)

Reviewer #2: Minor Revision

Reviewer #3: Acceptance

**Results**

-Does the analysis presented match the analysis plan?

-Are the results clearly and completely presented?

-Are the figures (Tables, Images) of sufficient quality for clarity?

Reviewer #1: (No Response)

Reviewer #2: Minor Revision

Reviewer #3: Acceptance

**Conclusions**

-Are the conclusions supported by the data presented?

-Are the limitations of analysis clearly described?

-Do the authors discuss how these data can be helpful to advance our understanding of the topic under study?

-Is public health relevance addressed?

Reviewer #1: (No Response)

Reviewer #2: Minor Revision

Reviewer #3: Acceptance

**Editorial and Data Presentation Modifications?**

Reviewer #1: (No Response)

Reviewer #2: Minor Revision

Reviewer #3: (No Response)

**Summary and General Comments**

Reviewer #1: The manuscript entitled “Seroepidemiologic survey of emerging vector-borne infections in South Korean forest/field workers” describes the sero-epidemiological feature of SFTSV and TBE. However, it didn't deviate much from showing predictable data. So, I recommend that authors would rather contribute to another suitable journal.

Reviewer #2: The manuscript submitted by Jiyun et al describes seroepidemiologic survey of emerging vector-borne infections in South Korean forest/field workers, and provides the evidence of SFTSV and some flaviviruses potential exposure. The manuscript should be published after revision. Some of the helpful suggestions and comments are below.

Please unify the positive rate on line 50…“0.1% (1/583)” and line 168…“0.2% (1/583)”, both for “0.2% (1/583)? or 0.17% in all the statement about SFTSV seropositivity in this paper. 

Line 50: If the seroprevalence including the positivity of neutralization, ELISA, and IFA, please re-describe it as TBEV and SFTSV were 1.03 % (6/583) and 0.17% (1/583).

Line 92 and 100: Please provide the strain or lineage information of TBEV and SFTSV.

Line 121-123: Author described that one forest worker in Hongcheon showed very low neutralization titer (1:10) against TBEV, also, anti-TBEV IgG was not detected using ELISA in this person”, is this serum also negative for TBEV IgM? Can't you be able do IFA with TBEV infected antigen slides? 

Line 184: Why the neutralization test for SFTSV was not performed? Because you don't have the SFTSV strain? If no, how you can able to prepare antigen slides infected with SFTSV in line 100? 

Line 320-321: Please clarify the additional details of positive human serum against SFTSV and normal human serum, where did the sera come from? How are those sera identified?

Table 2: The flaviviruses (TBEV, WNV, DENV, and JEV) were tested positive at the sampling places Gangneung and Jeju, Is there cross identification between them?

Table 2: The JEV was detected positive in all the sampling places, have those people ever been bitten by ticks? Are there higher positive rates of JEV carried by ticks in those areas?

Reviewer #3: Major

1. Authors describe “only 19 (3.3%) reported experience of tick bite” in line 113 and Seroprevalence of anti-TBEV IgG was 0.9% (5/583, 95% confidence interval [CI], 0.3-2.1) by ELISA. Neutralization assay for TBEV showed positive rate of 0.3% (2/583, 95% CI, 0.1- 118 1.4) between line 116 and line 118.

Are 19 (3.3%), experience of tick bite, have seropositive of anti-TBEV IgG or positive of neutralization assay for TBEV? 

The main transmission route of TBEV is tick bite. Therefore, could you describe about this in manuscript? 

2. Between line 124 and 124, authors describe below

“Among 583 subjects, positive rates for anti-DENV IgG and anti-WNV IgG were 0.5% (95% CI, 0.1-1.6). Peoples who have anti-DENV IgG or anti-WNV IgG also positive for anti- JEV IgG”

Antibodies of WNV have cross reaction with antibodies of JEV and gold standard of diagnosis of WNN is RT-PCR. 

Is positive rate for anti-WNV IgG real positive for WNV? 

3. In line 127, authors describe “Only 1 forest worker in Yeongju was seropositive for SFTSV by IFA (Fig 2)” and the title (1:2,048) is relative high. 

Could you do (real-time) RT-PCR about this specimen of this people? 

Because atypical signs and symptoms and asymptomatic infections have also been identified and reported. 

Minor

In line 33, Could you write full name of ELISA? 

In line 34, Could you write abbreviation of indirect immunofluorescence assay?

PLOS authors have the option to publish the peer review history of their article (what does this mean?). If published, this will include your full peer review and any attached files.

Reviewer #1: No

Reviewer #2: No

Reviewer #3: Yes: Keun Hwa Lee
---

## [Decision Letter · Decision Letter 1]

20 Jun 2021

Dear Dr. Song,

Thank you very much for your patience and submitting your manuscript "Seroepidemiologic survey of emerging vector-borne infections in South Korean forest/field workers" for consideration at PLOS Neglected Tropical Diseases. As with all papers reviewed by the journal, your manuscript was reviewed by members of the editorial board and by several independent reviewers. The reviewers appreciated the attention to an important topic. Based on the reviews, we are likely to accept this manuscript for publication, providing that you modify the manuscript according to the review recommendations. Especially, the authors need to revise the manuscript according to the comments suggested by reviewer #3. 

Sincerely,

Nam-Hyuk Cho

Deputy Editor

Nam-Hyuk Cho

Deputy Editor

Reviewer's Responses to Questions

**Key Review Criteria Required for Acceptance?**

**Methods**

-Are the objectives of the study clearly articulated with a clear testable hypothesis stated?

-Is the study design appropriate to address the stated objectives?

-Is the population clearly described and appropriate for the hypothesis being tested?

-Is the sample size sufficient to ensure adequate power to address the hypothesis being tested?

-Were correct statistical analysis used to support conclusions?

-Are there concerns about ethical or regulatory requirements being met?

Reviewer #1: (No Response)

Reviewer #3: YES

Reviewer #4: While information on the testing details for TBEV and SFTSV are provided, there is no mention of the other diseases listed in lines 68-70 in the Introduction. It is good to know that these other viruses were also tested, so I believe there should be a brief description about their tests in the Methods (please see my comments also below under Results and Conclusions). Also, the authors may wish to reconsider the title (i.e. “emerging vector-borne infections”), since the authors focus their paper on TBEV and SFTSV…

**Results**

-Does the analysis presented match the analysis plan?

-Are the results clearly and completely presented?

-Are the figures (Tables, Images) of sufficient quality for clarity?

Reviewer #1: (No Response)

Reviewer #3: YES

Reviewer #4: - As stated for the Introduction and Methods sections, a brief statement about the negative results for the other viruses tested from the specimens of all 583 persons would be informative. Table 2 is presented, but it does not seem to be referenced in the main text; for Table 2, what the numbers presented represent should be clearly indicated (i.e. “n (%)”; this should be listed at the top, not under TBEV). Some of the details regarding these tests become clear only in the Discussion section (lines 198-202, 212-213)...

- Table 1: Having this type of standard table 1 to show the demographic features of the study population is useful, but the presentation could be improved to help guide the reader. For instance, the years when these data were collected would be useful (2017-2018). In addition, the title heading (below “N=583”) could have “n (%)”, and an asterisk could be used to show where the data are “median (range)” (e.g., age, occupation duration, field worker duration). Also, for the Occupation category, it states that range is in “yr” but “1 month-50” appears, while the other says simply “1-60”, so this should also be standardized in presentation. And, form the text (lines 127-8), it appears that the “-“ were “0” (e.g., Vaccination history for yellow fever, past history of vector-borne diseases), so if that were the case, an explicit “0 (0)” is probably more appropriate. Also, for these data on vaccination histories and past history of diseases, if they were based on self-report, that should be explicitly indicated (as with the tick-bite experience).

- Table 1 and Results (or Discussion): albeit self-report, it would be useful to know what the results were for these 19 persons with a tick bite (for instance, were they more likely to have tested positive for SFTSV or TBEV?)—if there was anything notable about the 19, perhaps that could be briefly commented (conversely, if none of the seropositive SFTSV or TBEV individuals reported a tick bite, that would also be worth mentioning).

**Conclusions**

-Are the conclusions supported by the data presented?

-Are the limitations of analysis clearly described?

-Do the authors discuss how these data can be helpful to advance our understanding of the topic under study?

-Is public health relevance addressed?

Reviewer #1: (No Response)

Reviewer #3: YES

Reviewer #4: Line 169: as commented in the Introduction, can the authors be sure that the possible infection in China was asymptomatic (he may have had symptoms/signs but was not tested and diagnosed at the time)?

Line 177: as with the abstract, I would state as the number of “reported” cases since these are surveillance data.

Lines 222-224: while this is my personal view, I believe that the statement, “it is necessary to raise the awareness of physicians about TBEV infection and to establish a surveillance system in South Korea” is quite strong given what the study found and what the authors comment in their limitations. Perhaps additional special studies are needed before concluding that it is necessary to establish a surveillance system for TBEV (given the work necessary for setting up a public health surveillance system, including legal procedures, etc.)

**Editorial and Data Presentation Modifications?**

Reviewer #1: (No Response)

Reviewer #3: YES

Reviewer #4: Minor comments:

The English was competent, but there were some minor issues (phrasing, verb tense, etc.) so a review of the English by a native scientific writer would make the paper more readable. 

Some examples are listed below:

- Line 57: “Although existed long time ago”

- Line 107: “After washed”

- Line 138-139: “Peoples who have anti-DENV IgG or anti-WNV IgG also positive for anti- JEV IgG.”

- Line 190: “seropositive rate has decreased”

- Line 203: “one serum which is positive by”

- Line 354: "foreast"

**Summary and General Comments**

Reviewer #1: Although this data could not show direct the evidence of TBEV, I agree this study could attribute to development of vaccine policy. I had several questions about result, but other reviewer already has been requested. So, I have resolved by author's answers.

Reviewer #3: NOTHING

Reviewer #4: The original article, "Seroepidemiologic survey of emerging vector-borne infections in South Korean forest/field workers" (PNTD-D-21-00010R1), is a straightforward report regarding a serological investigation of tick-borne encephalitis virus (TBEV) and severe fever with thrombocytopenia syndrome virus (SFTSV) infections among forest and field workers in South Korea. The sample was large with nearly six hundred persons, and the seroprevalence for TBEV was 2/583 (0.3%) and 1/583 (0.2%) for SFTSV. The authors recommend raising awareness of physicians about TBEV infection and to establish a surveillance system for TBE in South Korea.

A few general comments not listed in the above sections:

Abstract, Line 27: “…the number of cases has increased…” should probably be stated as “…the number of reported cases has increased…”, since these are passive surveillance data that are reported by healthcare workers (which are dependent on patients seeking care, the clinician suspecting disease, testing for the disease, and ultimately reporting the disease). This applies to other areas where this phrasing appears (e.g., Author summary, Discussion).

Introduction, Line 71: while serologic surveys can be useful for detecting asymptomatic infections, given that the authors tested with IgG, I would think that past infections that went undetected/unreported would also be captured and offer valuable information. Do the authors really mean to limit their statement to "asymptomatic infections"? This applies to other areas where the emphasis is made on asymptomatic infections.

PLOS authors have the option to publish the peer review history of their article (what does this mean?). If published, this will include your full peer review and any attached files.

Reviewer #1: No

Reviewer #3: No

Reviewer #4: No

Figure Files:

Data Requirements:

Reproducibility:

References

---

## [Editor Report · Decision Letter 2]

27 Jul 2021

Dear Dr. Song,

We are pleased to inform you that your manuscript 'Seroepidemiologic survey of emerging vector-borne infections in South Korean forest/field workers' has been provisionally accepted for publication in PLOS Neglected Tropical Diseases.

Best regards,

Kendall McKenzie

Staff Admin

Nam-Hyuk Cho

Deputy Editor

---

## [Editor Report · Acceptance letter]

13 Aug 2021

Dear Dr. Song,

We are delighted to inform you that your manuscript, "Seroepidemiologic survey of emerging vector-borne infections in South Korean forest/field workers," has been formally accepted for publication in PLOS Neglected Tropical Diseases.

Best regards,

Shaden Kamhawi

co-Editor-in-Chief

Paul Brindley

co-Editor-in-Chief
